# Development of Sense of Coherence Stability in the AGORA Healthy Ageing Study

**DOI:** 10.3390/ijerph192114190

**Published:** 2022-10-30

**Authors:** Francesca Sanna, Maura Galletta, Maria Koelen, Paolo Contu

**Affiliations:** 1Department of Medical Sciences and Public Health, University of Cagliari, Blocco I, SS 554, km 4,500, 09042 Monserrato, Italy; 2Department of Social Sciences, Wageningen University & Research, 6700 EW Wageningen, The Netherlands

**Keywords:** sense of coherence, adulthood, Netherlands, Antonovsky, health, life events

## Abstract

Sense of coherence (SOC) is a psycho-social trait formed in childhood or adolescence, allowing individuals to be more resilient to daily life stressors, stay well, and improve their personal health. Although SOC remains stable after the age of thirty, only a few studies investigated its stability in adulthood. The aim was to investigate the development of SOC over time in 489 participants and its association with age, gender, educational level, or negative life events. The study was performed as part of the Healthy Ageing project of the Academic Collaborative Centre AGORA, a longitudinal study involving four municipalities of Eastern Netherlands. A self-administrated questionnaire was used to monitor the SOC of the elderly in 2008, 2010, and 2013, using the Orientation to Life Questionnaire (SOC-13). The analysis included repeated-measures ANOVA analysis and bivariate analysis using Pearson’s chi square test. We found no statistically significant variation in SOC over time (F (2, 282) = 2.99, *p* = 0.052) and no significant association with age (F (2, 282) = 2.851, *p* = 0.06), gender (F (2, 282) = 0.845, *p* = 0.43), or educational level (F (2, 282) = 0.708, *p* = 0.49). SOC remained stable in the elderly population, even if they experienced negative events over their lifespan.

## 1. Introduction

Across the world, people are increasingly living longer. By 2050, people aged 60 years and older are expected to total 2 billion [1]. The increased lifespan of populations leads to one of the most important challenges for public health, namely, successful ageing. The World Health Organization defines healthy ageing as “the process of developing and maintaining the functional ability that enables wellbeing in older age” [1]. Hansen-Kyle summarized it as a process of physical and cognitive slowing down, while resiliently adapting and compensating to optimally function and participate in all areas of one’s life, such as physical, cognitive, social, and spiritual [2] (p. 52). In 1948, the WHO defined health as “a state of complete physical, mental, and social well-being and not merely the absence of disease or infirmity”. This definition has become a discussion topic at the international level. A point of criticism concerns the absoluteness of the word “state” because health is a process [3,4]. Various studies about the perceptions of older people showed the existence of a link between old age, a positive sense of well-being, and mental health [5]. This association can be explained by the sense of coherence (SOC). Antonovsky [6] first introduced the concept of SOC as a dispositional orientation that allows individuals to be more resilient to daily life stressors, stay well, and improve their personal health. For instance, SOC contributes to health outcomes by reducing the impact of stress on physiological functioning, as well as mediating the relationship between stressful circumstances and well-being outcomes such as psychosomatic symptoms and poor self-rated health [7]. According to Antonovsky [3,4], SOC is a psychosocial trait formed in childhood or adolescence that remains generally stable after the age of thirty. A high SOC protects people from stress by the way they perceive life events as challenges, as occurring for a reason, and that, even if not under their own control, they can be handled by some other available resources [8]. The hypothesis of Antonovsky has remained largely untested in the general population, especially over the second half of a life span. In agreement with some studies, SOC increases as people get older instead of stabilizing [9,10]. Eriksson et al. [11] described the distribution of the SOC in people aged between 40 and 70 years, showing how it increased with age. Older age groups scored higher on SOC than younger groups, although the differences were small. Further, Lindmark et al. [12], using a Swedish version of the 13-item scale, in their study found that SOC increased with age and that young adults aged between 20 and 29 years had a statistically and significant lower SOC score than individuals older than 30 years. On the other hand, a Sweden longitudinal study conducted among adults from 55 to 101 years of age showed that SOC decreases with advancing age and that personal resources, such as having a higher level of education, moderate its decline though a life span [13]. Using a 13-item version of the SOC scale, in their prospective and longitudinal study among 56 participants, Lövheim et al. [14] reported a significant negative correlation between negative life events and the change in SOC scores over a five year period, meaning that the more negative life events there were, the more SOC decreased. Moreover, the authors found that one participant had an increase in SOC, even though he had experienced a high number of negative life events. Antonovsky never expressed which level of SOC is considered as normal and he recommended to analyse the concept of SOC as a whole entity, without dividing the sum of items into low or high. However, a number of studies reported division into low, moderate, or high SOC [15]. A high SOC suggests that an individual owns resources that enable the person to cope with stressful life events. Although changes in the SOC during a life course has been investigated in diverse studies, the results are controversial, and little is known about how SOC develops over the second half of life, as age-related deficits accumulate. We hypothesized that the development of a SOC over time in late adulthood would be associated with age, gender, educational level, or negative life events. 

## 2. Materials and Methods

### 2.1. Participants and Data Collection

For this study, we used existing data from the Academic Collaborative Centre AGORA Healthy Ageing study, a longitudinal cohort study with three data-point measurements, conducted in three municipalities of Eastern Netherlands (Epe, Zutphen, and Apeldoorn). 

Baseline measurements took place in a 10-week period in 2008. The follow-up measurements were in 2010 and 2013. A standardized, self-administrated questionnaire (“Elderly monitor 2008”) was developed in order to monitor the health status of the elderly. The questionnaire was pretested in a group of five voluntary elderly advisors to evaluate the acceptability of the questions by the study population and the applicability for self-administration. Although the survey is part of a larger study on the quality of life of the elderly, for the purpose of this study we used the SOC measure, demographic information such as gender, age, and education, and information about life events. Between 2008 and 2013, *n* = 5648 participants responded to the questionnaire, while *n* = 489 participants participated at all the three years of follow up. 

### 2.2. Measures

*Sense of Coherence*. This variable was measured by using the Dutch version (13 items) of the Orientation to Life Questionnaire (SOC-13). The questionnaire was valid and reliable with a Cronbach’s Alpha of 0.83 and it consists of three components: Comprehensibility (5 items), Manageability (4 items), and Meaningfulness (4 items). Comprehensibility refers to the ability of people to recognize their situation as understandable or practicable. Manageability refers to the perception of people of their capability to cope with difficult situations. Meaningfulness is a motivational dimension and measures the ability to find meaning in everyday events. The SOC score ranges between 13 and 91 points. According to Eriksson [16], a SOC score of 13–63 corresponds to a low SOC, 64–70 points to moderate SOC, and 80–91 to a high SOC.

*Life events.* Binary questions (yes, no) related to life events in the previous 12 months of the interview were selected from the main questionnaire. We identified 9 single items on negative events that we included in the descriptive analysis. The following items were included in the analysis: “Did you experience severe illness?”; “Did you divorce or break up with your partner?”; “Did you experience severe illness of your partner or family member?”; “Did you experience the admission of the partner in nursing home or in home for elderly?”; “Did you experience the death of your partner?”; “Did you experience the death of a close relative different from the partner?”; “Did you experience an important degradation of financial situation?”; “Did you divorce or break up with the partner?”, “Did you experience a severe conflict?”; “Did you experience any other radical events?”.

### 2.3. Study Setting

The study was performed in the frame of the AGORA Healthy Ageing study. In 2005, six community health services in the Netherlands collaborated to monitor the health status, health determinants, and use and need for health care of the elderly (aged 65 and older). Data were collected in the same period in all the community health services through standardized questionnaires with validated questions. Since this study was not invasive to the participant’s integrity, it did not require a formal ethics review following the criteria of the Medical Research Involving Human Subjects Act. A questionnaire was mailed to the general population, randomly sampled from the Municipal Database. Participants signed the informed consent to participate in the study [17,18]. They were also guaranteed of confidentially of their information and were notified that participation would be voluntary. Questionnaires were anonymous and all data were stored only using an anonymous ID for each participant, allowing the researchers to merge the longitudinal data in one dataset. 

### 2.4. Statistical Analysis

Negatively keyed items of SOC-13 were reverse recoded to make all variables positively keyed. Subjects with more than two missing values for each group of questions mentioned above were not considered for the analysis. Repeated measures ANOVA analysis was carried out to analyse the development of SOC in adulthood. The normality of the data was determined prior to performing repeated-measures ANOVA and data were checked for sphericity. When measuring change in the SOC score, we used the time as independent variable. Each level (or related group) represented a specific time point of the follow-up (2008, 2010, and 2013). The within-subject variable was represented by the SOC score of each subject for the three measurements. Age, gender, and educational level were included to analyse their association with SOC. These covariates were considered as a between-subject factor as we considered the SOC across these groups. To explore individual SOC in the participants, a cross table on the percentile distribution was created for the three waves. The row differences in SOC score between the second and first wave and between the third and second wave were measured for each individual of the study population. Then, the percentiles of the differences were calculated, and the participants were divided in seven groups based on the assigned percentile. Negative values labelled as −3, −2, and −1 were attributed to participants with a decrease in SOC score, and positive values labelled as +1, +2, and +3 were attributed to participants with an increase in SOC score. A null value was attributed to participants whose SOC had been stable throughout the two measurements. Once the categories were identified, an investigation on negative life events was conducted for the participants. Descriptive analysis was performed, and bivariate analysis was conducted using Pearson’s chi square test. The data were analysed using the statistical package IBM SPSS Statistics, version 20.0. The level of significance was set at the recommended 0.05 level. 

## 3. Results

In total, 489 subjects (237 males, 252 females, mean age at baseline ± SD = 72.6 ± 5.4 years) were included in the analysis. The age of the participants at baseline (2008) was between 70 and 97 years. The characteristics of the study population are shown in Table 1. The analysis of SOC in the sample population showed a mean score of 69.2 in both 2008 and 2010, and 68.2 in 2013. Mauchly’s Test of Sphericity indicated that the assumption of sphericity was not violated (χ^2^(1) = 0.496, *p* = 0.780). The results from the repeated-measures ANOVA showed that there was a not statistically significant variation in SOC over time (F (2, 282) = 2.99, *p* = 0.052). SOC was not associated with age (F (2, 282) = 2.851, *p* = 0.06), gender (F (2,282) = 0.845, *p* = 0.43), or educational level (F (2, 282) = 0.708, *p* = 0.49). Almost 70% (*n* = 345) of the participants did not present significant variation over time, of which 34.5% (*n* = 169) of the whole study population had a complete stable SOC during the follow up. A total of 8.1% (*n* = 40) of the participants had a decrease in SOC between the first and the second wave, followed by an increase between the second and the third wave (Table 2). About 11% (*n* = 52) of the participants showed an increase in SOC between the first and the second wave, followed by a decrease between the second and the third wave. A total of 3% (*n* = 15) of the participants had continuously increased or decreased their SOC during the study period. Specifically, *n* = 7 had a double decrease in SOC through the three measurements, and *n* = 8 of the participants had a double increase through the waves. All the remaining participants had a stable SOC between 2008 and 2010 or between 2010 and 2013.

The prevalence of participants with a decreased SOC over time and who experienced negative life events in the prior 12 months of follow-up was lower than 16%, as shown in Appendix A.

## 4. Discussion

The aim of the present study was to describe the development of SOC over time in an elderly cohort and to investigate its association with age, gender, educational level, and negative life events. Questions about how SOC changes over time have been addressed in a few previous studies [8,9,11,12]. However, relatively little attention has been dedicated to the development of SOC over the second half of life. Findings are controversial and it is still unclear whether the SOC increases, decreases, or remains stable as people get older. In our study population, SOC was stable during the three years of follow up in 2008, 2010, and 2013. Specifically, the mean SOC score was 69.2 in 2008 and 2010, with a decrease of 1.1 point in 2013, reaching a mean score of 68.1. Although various researchers reported a continuous increase in the SOC into advanced old age [8,9], this study showed SOC stability in adulthood, supporting Antonovsky’s original assumption. Our findings are also in line with a Swedish study in which no differences were found in SOC between home-dwelling individuals and patients [19]. Moreover, we found that age, gender, and educational level were not associated with SOC development in late adulthood. In agreement with our results, studies reported no differences between males and females in the SOC of an adult [19] general population [9]. On the other hand, some studies have found differences in gender, reporting men with a significantly higher SOC score than women, specifically in response to severe life events [20].

Some other studies [21,22,23] found that educational level, mental health, and quality of life were a significant determinant of SOC [24]. In fact, the higher the level of education the older adult had achieved, the higher the SOC. However, differently from our study, they measured SOC using the Spanish version of a shortened version of Antonovsky’s scale.

We also aimed to explore the negative life events experienced by elderly and we have seen that negative events seem to not have a relevant influence on the increase or decrease in the participants’ SOC over time, nor in those with a constant decrease in SOC during the whole period. Previous research found a negative correlation between negative life events and the change in SOC score; thus, more negative life events decrease the SOC [14]. The ageing process is often characterized by many physical changes and is generally thought to be a stressor that can create the feeling that everyday life is difficult to manage. However, from our study it emerges that the prevalence of participants who experienced negative life events and a decrease in SOC over time was low. Indeed, it might be possible that negative life events also function as a learning life experience that have strengthened the SOC of the participant. The results must be treated carefully since the statistical analysis performed through the repeated-measures ANOVA considers the variability due to individual differences but does not explain the individual change in SOC. 

A number of limitations provide directions for future research. Firstly, although the analysis did not allow us to explain the individual change in SOC based on the participant’s characteristics, we designed a cross table to have a descriptive representation of the individual SOC trends. Secondly, life events were defined as negative by the researchers and whether a life event is negative or positive depends on how it is perceived by the individual. Future development of a study that could integrate positive life events, such as marriage, birth of child, etc., would be beneficial to analyse if they contribute to increasing SOC. 

We investigated the negative life events for each participant, according to their change in SOC over time, and we analysed how the SOC of participants increased, decreased, or stayed stable through adulthood, capturing the individual direction of change. The majority of the study population had a stable SOC from baseline to follow-up. Although the temporal space differs between the years of follow-up (2008–2010, 2010–2013), our results showed some individual variation as people get older. Minority groups of participants were identified, including those with a decrease in SOC during the first year of follow-up, followed by an increase in the third year of follow-up; those with an increase in SOC during the first year of follow-up, followed by a decrease during the third year of follow-up; and those who had a continuous increase or decrease in SOC over time. 

The results suggest that SOC might be related to other measures, for example, to good health or mental health [16,25,26]. Recent research on the development of the SOC suggested future directions to develop salutogenic research, encouraging alternative approaches to measuring the SOC and prioritizing a better understanding of its origins in early life [27].

## 5. Conclusions

To the best of our knowledge, this is the first longitudinal study investigating the variation in sense of coherence in an older Dutch population. The study provides a significant contribution to research, showing a stability in SOC over a five-year period, during a phase of life rarely investigated. No association was found between the development of SOC and negative life events experienced by participants during the previous 12 months of follow up. Furthermore, no association was found between the development of a SOC over time and age, gender, and educational level. 

Because more people live to an old age, the burden of disease will also increase. Moreover, as people get older, they have new challenges and fewer resources to cope with their life. Thus, it is crucial for professionals and policymakers to focus on strengthening the physical and mental capacity of the elderly and creating an environment to allow them to achieve their goals. Future research efforts should include work investigating the relationship between SOC and events that occurred in life. Indeed, it is crucial to support people to deal with everyday life challenges and encourage them to be engaged in everyday life activities that are meaningful to them. To conclude, further research on SOC might offer the opportunity to disentangle the mechanism behind its development and help implement ad-hoc strategies to facilitate coping in older age.

## Figures and Tables

**Table 1 ijerph-19-14190-t001:** Characteristics of the study population according to gender.

	Male (*n* = 237)	Female (*n* = 251)	Total	*p* Value
**Age at baseline, years**	72.12 (±5.41)	73.04 (±5.34)	72.58	0.464
**Sex, %**	237 (48.57)	251 (51.43)	488	<0.001
**SOC**				
2008	70.04 (±9.83)	68.34 (±10.3)	69.19	0.554
2010	69.57 (±9.918)	68.8 (±9.19)	69.19	0.772
2013	68.81 (±9.65)	67.57 (±10.10)	68.18	0.293
**Education level at baseline, %**				<0.001
No/Primary	19 (8.26)	44 (18.18)	63	
Low	85 (36.96)	135 (55.79)	220	
Medium	47 (20.43)	33 (13.64)	80	
High	79 (34.35)	30 (12.40)	109	
	**NEGATIVE LIFE EVENTS**			
	**Severe illness of oneself, %**			
2008	19 (8.92)	17 (7.91)	36	<0.001
2010	26 (12.21)	18 (8.41)	44	0.013
2013	27 (12.50)	26 (12.56)	53	0.182
	**Severe illness of partner or family member, %**			
2008	18 (8.49)	26 (11.87)	44	<0.001
2010	33 (14.73)	27 (12.05)	60	0.002
2013	32 (14.95)	18 (8.78)	50	0.023
	**Admission of the partner in nursing home or in home for the elderly, %**			
2008	1 (0.47)	3 (1.38)	4	<0.001
2010	5 (2.25)	3 (1.35)	8	<0.001
2013	1 (0.47)	4 (1.96)	5	0.018
	**Death of the partner, %**			
2008	3 (1.42)	5 (2.31)	8	<0.001
2010	3 (1.36)	9 (3.98)	12	<0.001
2013	7 (3.32)	8 (3.92)	15	0.102
	**Death of a close relative different from partner, %**			
2008	34 (16.11)	33 (15.07)	67	<0.001
2010	40 (18.10)	39 (17.18)	79	0.015
2013	26 (12.26)	35 (16.67)	61	0.172
	**Important degradation of financial situation, %**			
2008	11 (4.80)	12 (5.00)	23	0.913
2010	14 (6.03)	11 (4.62)	25	0.502
2013	18 (8.41)	20 (9.90)	38	0.441
	**Divorce or break up with partner, %**			
2008	1 (0.47)	0 (0.00)	1	<0.001
2010	1 (0.44)	0 (0.00)	1	<0.001
2013	0	0	0	<0.001
	**Flaming row or severe conflict, %**			
2008	6 (2.62)	8 (3.35)	14	0.639
2010	3 (1.31)	3 (1.27)	6	0.970
2013	3 (1.40)	8 (3.86)	11	0.217
	**Other radical events, %**			
2008	8 (3.77)	10 (4.69)	18	0.630
2010	6 (2.82)	12 (5.48)	18	0.163
2013	11 (5.39)	8 (4.32)	19	0.479

All values are given as the mean ± standard deviation (SD), unless otherwise stated. Abbreviation: SD, standard deviation; SOC, sense of coherence.

**Table 2 ijerph-19-14190-t002:** Percentiles distribution of SOC among the participants during the years of follow up.

			SOC Score between 2010 and 2013							
			Decrease			Stability	Increase			
			**−3**	**−2**	**−1**	**0**	**1**	**2**	**3**	**Total**
**SOC score between 2008 and 2010**	**Decrease**	**−3**	0	0	0	8	0	2	4	14
		**−2**	0	0	1	5	3	2	4	15
		**−1**	0	1	5	40	14	9	2	71
	**Stability**	**0**	6	7	43	169	30	14	4	273
	**Increase**	**1**	1	4	15	26	3	0	0	49
		**2**	5	2	6	16	2	0	2	33
		**3**	8	3	8	14	1	0	0	34
		**Total**	**20**	**17**	**78**	**278**	**53**	**27**	**16**	**489**

Percentiles are expressed as a decrease (−3, −2, −1), increase (1, 2, 3), or stabilization (0) of the SOC during the three waves of follow up.

## Data Availability

The data presented in this study are available on request from the corresponding author and first author.

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
