# Peer review of "Development of Sense of Coherence Stability in the AGORA Healthy Ageing Study"

_ijerph, 2022, doi:10.3390/ijerph192114190_

Round 1

Reviewer 1 Report

The paper researches about Sense of coherence (SOC) stability in adulthood, namely its association with age, gender, educational level, or negative life. The authors conclude that SOC remained stable in the elderly population, even if they experienced negative events over the lifespan. For example, they refer that there is no  statistically significant variation in SOC over time with p= 0.052 and no significant association with age with p= 0.06. However, the p-values are very low, so the conclusion is too forced. This aspect is important, especilaly because the literature presents other studies with different results. In addition, the authors refer a longitudinal study regarding a 3-year period, which I think it is not enough for the study they intend to present. Therefore, the time period should be extended.

In my opinion, this article is just another study on sense of coherence in which the authors conclude that there is no significance regarding the variables involved. I consider that there are no relevant and innovative contributions to the literature. In addition, there is not enough information about the questionnaires that were not validated.

Acronyms in the abstract such as “AGORA” should be avoided. In lines 64-65, there are no citation sustaining “low, moderate, or high SOC.” Moreover, the conclusions are inadequate in relation to the study carried out.

I will end the review with several questions for reflection on the relevance of the article:

What is the importance of investigating the development of SOC over time and its association with age, gender, educational level, or negative life?

What are the implications of this study for research? What are the new contributions of the study to the literature?

What methods were used in the other studies presented in the literature review? More information about the previous studies, namely about the methods and tools used, could provide more information to research, namely if different methods can conduct to different results.

Author Response

Dear Editor,

Thank you for giving me the opportunity to submit a revised manuscript titled “Development of sense of coherence stability in the AGORA Healthy Ageing study” to the IJERPH Journal. We appreciate the time and effort that you and the reviewers have dedicated to providing your valuable feedback on the manuscript.

We are grateful to the reviewers for their insightful comments on the paper. We have been able to incorporate changes to reflect most of the suggestions provided by the reviewers. We have highlighted the changes within the manuscript.

Here is a point-by-point response to the reviewers’ comments and concerns. Lines are referring to the revised copy of the tracked manuscript.

Reviewer 1: The paper researches about Sense of coherence (SOC) stability in adulthood, namely its association with age, gender, educational level, or negative life. The authors conclude that SOC remained stable in the elderly population, even if they experienced negative events over the lifespan. For example, they refer that there is no statistically significant variation in SOC over time with p= 0.052 and no significant association with age with p= 0.06. However, the p-values are very low, so the conclusion is too forced. This aspect is important, especilaly because the literature presents other studies with different results. In addition, the authors refer a longitudinal study regarding a 3-year period, which I think it is not enough for the study they intend to present. Therefore, the time period should be extended. In my opinion, this article is just another study on sense of coherence in which the authors conclude that there is no significance regarding the variables involved. I consider that there are no relevant and innovative contributions to the literature. In addition, there is not enough information about the questionnaires that were not validated.

C1. Acronyms in the abstract such as “AGORA” should be avoided.

R1. Thank you for the suggestion. We agree we should avoid acronyms in the abstract. However, the terms AGORA in this case indicates the entire name of the Academic Collaborative Centre located in the Netherlands.  

C2. In lines 64-65, there are no citation sustaining “low, moderate, or high SOC.” Moreover, the conclusions are inadequate in relation to the study carried out.

R2. Thank you for this observation. We added the reference of the review carried out by Eriksson et al. (14) in which they investigated published studies that used sense of coherence divided into low to high.  

I will end the review with several questions for reflection on the relevance of the article:

C3. What is the importance of investigating the development of SOC over time and its association with age, gender, educational level, or negative life?

R3. Sense of coherence is an important dispositional orientation that allows individuals to be more resilient to daily life stressors, stay well and improve their health. Life-expectancy rates   worldwide is increasing resulting in a progressive increase of the elderly population. Thus, it is fundamental to understand how SOC develops in this part of the population and understand the mechanisms and resources from which individuals can strengthen their health.

C4. What are the implications of this study for research? What are the new contributions of the study to the literature?

R4. Sense of coherence is a psychosocial trait formed during young age. Its development has been investigated on several studies. Although Antonovsky stated SOC remains stable after adulthood, this hypothesis it has not been largely proved and it remains unclear if SOC continues to develop into older age. The new contribution of this study is to examine the development of sense of coherence in an elderly population and provide contributions to literature investigating life events on the population.

C5. What methods were used in the other studies presented in the literature review? More information about the previous studies, namely about the methods and tools used, could provide more information to research, namely if different methods can conduct to different results.

R5. Information on methods and tool used in previous studies have been added in lines 52, 57, page 2.

Reviewer 2 Report

- Abstract - line 14: Are the terms "negative life" the most appropriate? Negative life events? cf. line 58

- Introduction - line 45. You must add a "." after the 7 in brackets. Line 49, also add a "." after Eriksson et al. Idem ligne 52, 58,...

> Language editing should be performed.

- Introduction. Overall, I find that this introduction could be more thorough concerning the SOC and its determinants, consequences, etc.

- Introduction. I think that the authors should more justify why they focus on people age 65 and more (cf. methods).

- Introduction. The authors describe their aims but they do not write any hypothesis. I think this should be added.

- Measures. Concerning "life events", how did the participants respond? Was it a "yes or no" answer or a more qualitative response? This should be more detailed.

- Measures; All measures should be described and not only the SOC and the life events. What about the perceived health, the wellbeing,...?

- Statistical analysis. Did you perform some statistical correction (e.g., Bonferroni ?)

- Results. Line 152. The mean of SOC in 2008 and 2010 was exactly the same?

- Results. Table 1. A column presenting the results for the whole sample should be added.

- Results. Why is the BMI included? This has not been discussed before. It should removed or discussed in the introduction. Also, why is the mental health, quality of life, etc. in relation to SOC not included? I think that this could enrich the discussion of the results. 

- Discussion. The temporal space between the measures are not the same (2 years between 2008 and 2010, 3 years between 2010 and 2013). This should be discussed. 

- Discussion. I do not understand why the version of the questionnaire can influence the results. Lines 195-199. Could you explain more thoroughly your point of view? 

- Discussion. Could you run the analyses without the two items (cf. line 223)? 

- Discussion. Regarding the emotional valence of an event, can you imagine another way to assess? For instance, you could use visual analogic scales to assess the valence of the emotion related to an event rather than assume that this kind of event is positive or negative for everyone...

- Discussion. I wonder if it would be relevant to integrate notions of autobiographical memory in relation to SOC? 

- Discussion. What are the clinical perspective of your results?

Author Response

Dear Editor,

Thank you for giving me the opportunity to submit a revised manuscript titled “Development of sense of coherence stability in the AGORA Healthy Ageing study” to the IJERPH Journal. We appreciate the time and effort that you and the reviewers have dedicated to providing your valuable feedback on the manuscript.

We are grateful to the reviewers for their insightful comments on the paper. We have been able to incorporate changes to reflect most of the suggestions provided by the reviewers. We have highlighted the changes within the manuscript.

Here is a point-by-point response to the reviewers’ comments and concerns. Lines are referring to the revised copy of the tracked manuscript.

Reviewer 2:

C6. Abstract - line 14: Are the terms "negative life" the most appropriate? Negative life events? cf. line 58

R6. Thank you for the comment. We revised the sentence as follow: “The aim was to investigate the development of SOC over time in 489 participants and its association with age, gender, educational level, or negative life events.”

C7. Introduction - line 45. You must add a "." after the 7 in brackets. Line 49, also add a "." after Eriksson et al. Idem ligne 52, 58,...

R7. The correction was made as suggested.

C8. Language editing should be performed.

R8. The language editing was made as required.

C9. Introduction. Overall, I find that this introduction could be more thorough concerning the SOC and its determinants, consequences, etc.

R9. Thank you for this comment. We added information on SOC in the introduction section with the following sentence “A high SOC protects people from stress by the way they perceive life events as challenges, occurring for a reason and that, even if not under their own control, they can be handled by some other available resources”, lines 46-49, page 2

C10. Introduction. I think that the authors should more justify why they focus on people age 65 and more (cf. methods).

R10. Thank you for the comment. We included more information on the introduction section, lines 46-49, page 2 and in the discussion section, lines 25- 254, page 7.

C11. Introduction. The authors describe their aims but they do not write any hypothesis. I think this should be added.

R11. Thank you for pointing this out. We agree that a priori hypothesis was not clearly stated. Thus, we revised the introduction with the following sentence: We hypothesized that the development the SOC over time in late adulthood would be associated with age, gender, educational level, or negative life events” on the introduction session, lines 69-71, page 2.

C12. Measures. Concerning "life events", how did the participants respond? Was it a "yes or no" answer or a more qualitative response? This should be more detailed.

R12. Thank you for the comment. Information on the life events measure have been added on line 101, page 3.

C13. Measures; All measures should be described and not only the SOC and the life events. What about the perceived health, the wellbeing,...?

R13. Thank you for this comment. We agree that if we are mentioning the scales, then these should be described. However, to avoid burdening the reading and confusing the reader, we changed the sentence as follows: "Although the survey is part of a larger study on the quality of life of the elderly, for the study purpose we used SOC measure, demographic information such as gender, age, and education, and information about life events" (lines 89-92, page 2).

C14. Statistical analysis. Did you perform some statistical correction (e.g., Bonferroni ?)

R14. Thank you for the comment. We have carried out post-hoc checks using Bonferroni test, after running the repeated measure ANOVA analysis through SPSS. Post hoc analysis confirmed our results.

C15. Results. Line 152. The mean of SOC in 2008 and 2010 was exactly the same?

R15. Thank you for the comment. The mean SOC for the sample population was 69,19 in 2008 while 69,185 in 2010. Therefore, we decided to leave the approximated number in the text. 

C16. Results. Table 1. A column presenting the results for the whole sample should be added.

R16. Thank you for this comment. Results for the whole sample have been added to Table 1.

C17. Results. Why is the BMI included? This has not been discussed before. It should removed or discussed in the introduction. Also, why is the mental health, quality of life, etc. in relation to SOC not included? I think that this could enrich the discussion of the results. 

R17. Thank you for pointing this out. We removed BMI to the Table 1 as suggested, as it was not included in the introduction. Further, we agree with the reviewers that is important to discuss mental health and quality of life in relation to SOC. Accordingly, we added this sentence “Together with other studies [19, 20], some [21] found that educational level was a significant determinant of SOC, as well as mental health and quality of life” on the discussion, lines 203-205, page 6.

C18. Discussion. The temporal space between the measures are not the same (2 years between 2008 and 2010, 3 years between 2010 and 2013). This should be discussed. 

R18. Thank you for this observation. We have included this sentence “Although the temporal space differs between the years of follow-up (2008-2010, 2010-2013), our results showed some individual variations as people get older.” on the discussion session, lines 229-231, page 6.

C19. Discussion. I do not understand why the version of the questionnaire can influence the results. Lines 195-199. Could you explain more thoroughly your point of view? 

R19. Thank you for this meaningful comment. We agree that this sentence might be unclear and perhaps it makes little sense to motivate the limitation, as a shortened version of the questionnaire should not influence the results. Thus, we decided to remove the sentence to avoid confusion to the readers (lines 229-231).

C20. Discussion. Could you run the analyses without the two items (cf. line 223)? 

R20. As we delete the sentence in the previously reply, we believe that it is not necessary to run the analyses without the two scale items.

C21. Discussion. Regarding the emotional valence of an event, can you imagine another way to assess? For instance, you could use visual analogic scales to assess the valence of the emotion related to an event rather than assume that this kind of event is positive or negative for everyone...

R21. Thank you for this comment. The use of visual analogic scales on this context would be meaningful. However, this scale (or similar) was not available on our study, as we used existing data from the Academic Collaborative Centre AGORA Healthy Ageing study.

C22. Discussion. I wonder if it would be relevant to integrate notions of autobiographical memory in relation to SOC?

R22. Thank you for this meaningful comment. We agree autobiography memory might be relevant in relation to SOC, as it might include memories for events occurred in childhood. However, we believe this might be not relevant to include in our study, as we considered life events occurred in the previous 12 months of the interview.

C223. Discussion. What are the clinical perspective of your results?

R23. Thank you for this meaningful observation. We included the following sentence “Because more people live into old age, the burden of disease will also increase. Thus, it is crucial for professionals and policymakers to focus in strengthening the physical and mental capacity of the elderly and creating the environment to allow them to achieve their goals.” in the discussion section, lines 253- 256, page 7.
